# Co-Opetitive Strategy Optimization for Online Video Platforms with Multi-Homing Subscribers and Advertisers

**Jing Li [1], Shuying Gong [2],\* and Xing Li [3]**

1 School of Economics and Management, Shanghai University of Political Science and Law, Shanghai 201701, China
2 College of Business, Shanghai University of Finance and Economics, Shanghai 200437, China
3 Department of History, The Chinese University of Hong Kong, Hong Kong, China
\* Correspondence: gong.shuying@163.sufe.edu.cn

**Abstract:** In the two-sided market for online streaming content, the platform's co-opetitive strategy has been wildly discussed, where the platforms cooperate in sharing the broadcasting right of content and meanwhile compete for both subscribers and advertisers. Although platform co-opetition in practice can be easily captured, the impacts of cross-side network effects on pricing strategy are contingent upon the participation decision of both sides, including single-homing and multi-homing. Therefore, we examine the optimal co-opetitive strategy of duopoly platforms using a Hotelling model to capture user behaviors and investigate the equilibriums of pricing decisions and profits in three scenarios: single-single, multi-single, and multi-multi. The main findings are: (1) Advertisers choose multi-homing only when subscribers are also multi-homing, and the broadcasting cost is relatively low. (2) With single-homing advertisers, the primary broadcasting platform earns more profit than the re-broadcasting one. (3) With multi-homing advertisers, the primary broadcasting platform's profit increases with the broadcasting rights cost. (4) Platforms should focus on building strong cross-side network effects with multi-homing advertisers. Alternatively, they would be better off contracting with single-homing advertisers if the effects are relatively low.

**Keywords:** multi-homing; co-opetitive strategy; cross-side network effect; pricing decision

## 1. Introduction

With the widespread use of advanced internet technology and the reduction in social contact during the epidemic, online video platforms have evolved. For example, Netflix, one of the most popular online video platform enterprises in the United States, had more than 200 million paid subscribers in the fourth quarter of 2020, up from 100 million in 2017 [1]. In 2020, the Chinese online video platform iQiyi was expected to generate about 30 billion yuan in total yearly income, up from 5 billion yuan in 2016 [2]. Furthermore, new internet live streaming services such as TikTok have exploded in popularity around the world. As of October 2020, TikTok had over 2 billion smartphone downloads worldwide [3]. On the one hand, the massive profit margin encourages additional firms to enter the video content market and compete, which significantly decreases the cost of subscribers switching across platforms. On the other hand, the platforms can gain revenue from business enterprises who want to advertise products during the commercial break. Moreover, obtaining broadcasting rights to major events (e.g., sports events such as the World Cup and the Olympic Games) could entice more users to subscribe and make the platform more attractive to advertisers [4]. Therefore, the platforms connect the subscribers and advertisers directly, resulting in a two-sided market.

As the online platform ecosystem expands in popularity, the subscribers place a higher value on watching experiences, preferring live streaming over replay [5]. Evidence shows that Facebook live-streaming videos are ten times more likely to elicit comments than recorded videos, are watched three times longer than ordinary videos, and have grown

in popularity by over 330% since the rollout. With the development of internet-based social networks and more users joining online video platforms, real-time word-of-mouth becomes more effective in facilitating the bandwagon effect; that is, the interaction between two-sided users will be enhanced in the live-streaming content market. As a result, the network externality effect of platforms is significantly amplified in this case. To benefit from this effect, the content creators begin to embrace a co-opetitive strategy, in which a primary streaming platform possesses the broadcasting rights, and then its competitive platform purchases the right of re-broadcasting before vying for users with this platform [6]. For example, in 2018, China Central Television shared its re-broadcasting right of the FIFA World Cup with Migu (miguvideo.com, accessed on 1 March 2023), bringing a three-times increase in Migu's daily active users compared to 2017 [7]. Similarly, Netflix made a contract with Youku (a Chinese streaming video platform) to share the broadcasting right of Day and Night across 190 countries and regions [8]. The reason for live streaming content platforms pursuing co-opetition is that the cross-side network effect will be significantly enhanced by this strategy in a two-sided market [9]. Cross-side network effects refer to the impact that one side of the platform (in this case, the advertisers) has on the other side (the subscribers).

The co-opetitive strategy is also embraced by other industries. In China's credit card payment system, consumers can conduct transactions directly with commercial banks. Nonetheless, third-party online payment platforms, such as Alipay, are more popular, even though they must be authorized by commercial banks. In this case, the commercial banks and Alipay construct a co-opetitive relationship, which means they compete while also cooperating in the market. In platform co-opetition, users on both sides of the market, subscribers and advertisers, can either join one platform, termed single-homing, or simultaneously join both platforms, termed multi-homing. Take the streaming content market as an example; the users on both sides have two entry options, leading to four different structures: both sides single-homing, both sides partially multi-homing, and one side single-homing but another side partially multi-homing. In this study, we will not consider the case of single-multi subscribers and partially multi-homing advertisers. The reason here is that the advertisers prefer to join multi-platform only with multi-homing subscribers. Otherwise, they are more willing to advertise just to their target customers at a relatively lower cost.

Overall, co-opetition has become quite common in the live-streaming market. Especially for platform industries, it can be an efficient way to save costs and avoid duplication of effort. While for platform co-opetition, more users get on board with the enhanced network effects. Thus, combining various market situations with a co-opetitive strategy and investigating the optimal mode selection of platforms are quite important, especially in a two-sided market with partially multi-homing users. However, due to the complexity of making the optimal pricing strategy in a two-sided market with multi-homing subscribers and advertisers, only a few platforms can gain more profit. For the development of the platform ecosystem, we suggest that the platforms can implement the co-opetitive strategy after addressing the following issues: (1) What are the equilibrium two-sided pricing strategies for platforms in different scenarios considering users' multi-homing behaviors? (2) What is the impact of cross-side network effects on the market share (i.e., the number of users) in different participation scenarios? (3) How will the equilibrium profits of platforms change with the primary market drivers for live streaming, such as network effect and broadcasting right cost?

To answer the above questions, in this paper, we focus on the co-opetitive strategies of two platforms in the streaming content market and meanwhile consider the participation decisions of both sides of users. First, we utilize the extended Hotelling model to characterize the user's behavior and then calculate the equilibrium numbers of users on different platforms. Second, we use the Bertrand duopoly model to optimize the pricing decisions for the broadcasting and re-broadcasting platforms. Based on the context, we investigate the equilibrium results of pricing, market share and platforms' profits in

three cases: single-homing subscribers and advertisers (as single-single), multi-homing subscribers and single-homing advertisers (as multi-single), and multi-homing subscribers and advertisers (as multi-multi).

According to the results, on the one hand, we find that the advertisers are more willing to be multi-homing with the multi-homing subscribers when the platform's broadcasting rights cost is relatively low. Furthermore, the advertisers prefer to join the primary broadcasting platform rather than the re-broadcasting one when the subscribers are single-homing. On the other hand, the primary broadcasting platform's profit is always higher than the re-broadcasting platform when the advertisers are single-homing. However, when the single-homing advertisers switch to multi-homing, the primary broadcasting platform's profit will increase as the broadcasting rights cost gets relatively higher. In the case of single-homing advertisers and subscribers, the primary broadcasting platform prefers to authorize more content to the re-broadcasting platform than in the case of multi-homing subscribers. In addition, we suggest that platforms cooperate with single-homing advertisers when the cross-side network effects are relatively low or, otherwise, choose the multi-homing advertisers when the network effects are amplified.

The remainder of this study is organized as follows. Section 2 examines the relevant literature and compares our paper to it. Section 3 introduces the basic model and solves three cases for optimal pricing decisions. Section 4 contrasts the optimal results in three different scenarios. The numerical analysis in Section 5 verifies the optimal results. Section 6 summarizes the conclusions. All proofs of the results are relegated to the Appendix A.

## 2. Related Literature

Three relevant streams of research have addressed the pricing strategies of a two-sided market, the online video platform and the platform co-opetitive mode.

### 2.1. Two-Sided Market and Pricing Strategy

Regarding the two-sided market and related pricing strategies, the pioneering works of Rochet and Tirole [10], Armstrong [4] and Eisenmann et al. [11] have provided the basic analytical framework for the pricing strategies under a two-sided market; subsequent studies have extended under both the same-side and cross-side network effects [11–13]. Under the two-sided market framework, existing literature has mostly shed light on the optimal pricing decisions and competitive strategies of platforms. For example, Barros et al. [14] offer two revenue-generating modes of media platforms, consumers subscribing to online video programs and advertising strategies. Fan et al. [15] indicate the optimal strategies for providers under the trade-off between the quality of the program and the cost for users to access it. Kind et al. [16] analyze how competitive force (number of media platforms) and media content differentiation influence the platform's revenue. Reisinger [17] examines a two-sided market where platforms compete for advertisers and consumers by considering the user's heterogeneity, suggesting that the platforms obtain positive margins in advertising. Gal-Or et al. [18] consider both the user-side and the advertiser-side's heterogeneity and suggest that advertising strategies depend on the users' heterogeneity.

Subsequently, the competitive mode unveils the phenomenon of muti-homing in the market [19]. For instance, Choi [20] finds that consumers' multi-homing can be beneficial to content providers and welfare enhancing. Ambrus et al. [21] indicate a new model for media platform competition and allow the users to subscribe to multiple platforms; they focus on the effects of advertising and consumers' entry and merge behavior. Anderson et al. [22] consider consumer multi-homing and advertising finance under the framework of the classic circle model, which analyzes the price changes under consumer multi-homing. D'Annunzio and Russo [23] consider two publishers and multi-home consumers and advertisers and study the impact of advertising networks on publishers and advertisers.

### 2.2. Online Video Platform

The second line of related literature considers the online video platform. Under the two-sided market framework, several papers consider different contexts, such as the ride-sharing industry, payment card industry, e-commerce industry, peer-to-peer lending industry, game industry and crowdfunding industry [9,24–28].

Different from these general two-sided platforms, online video platform owns two unique characteristics. First, an online video platform's strategy choice is an inherent trade-off [29], which means the increase in subscribers on the platform will lead to more advertisers' entry. However, a larger number of advertisers will decrease the number of subscribers [6,30,31]. Second, different from other two-sided market platforms, online video platform earns their profit from not only from users on two-sided (subscribers and advertisers) but also the platform content suppliers and the broadcasting cost [29]. For instance, classical and popular content is live streaming content such as the World Cup, the Olympic Games and the League of Legends Pro League. Compared with traditional recorded video, live streaming content enables users to interact closer with each other through the internet in a specific and fixed time [32]. Users during the live may have more followers and discussions, deeper engagements, and closer social ties, and thereby platforms will monetize heavier online traffic through advertising [33]. Thus, platforms may prefer to use the co-opetitive strategy and purchase the right of re-broadcasting before vying for subscribers with this platform [6]. Thus far, the existing literature related to online video platforms has mostly focused on strategies from two-sided users to maximize profit; few have paid attention to the cost of streaming content. Inspired by the existing literature and the specific context of live streaming on online video platforms in practice, we will analyze the equilibrium and streaming content pricing strategies under the network effects and two-sided online video platforms.

### 2.3. Co-Opetitive Strategy

A third strand of the literature is related to the co-opetitive mode in the platforms. The concept of "co-opetitive" was first suggested by Brandenbuger and Nalebuff [34], who indicated that firms cooperate with a competitor to achieve a common goal or get ahead. As we discussed above, the co-opetitive indicates the phenomenon of two competitors cooperating [28,35]. Compared with mere co-operation, the co-opetitive structure will also compete in the market. Co-operation is an overall win-win, but splitting the gains is a zero-sum game. Moreover, recently co-opetitive structure is common in a wide range of industries, such as Apple and Samsung, Ford and GM and Google and Yahoo [36]. Thus far, the related literature in Marketing and Operations Management could be divided into three parts: co-operation between different traditional markets [37–41], co-operation between platform firms and traditional markets [9,42–44], and co-operation between different platform firms [28; 45]. In specific, for the phenomenon of different traditional markets' co-opetitive, Casadesus-Masanell and Yoffie [37] analyze the competitive interactions between Inter and Microsoft, two complementary products, and demonstrate the R&D investment, pricing, release time, and value at different phases of product generations. Wright et al. [38] analyze the phenomenon of airlines' co-opetitive, in which flights operated by two or more airlines are sold together; the authors formulate a model of a two-partner alliance to calculate the effects on partners' behaviors. Niu et al. [39], motivated by Google's technology specifications on Android devices, analyze firms' decisions on production timing, ex-ante production strategy and ex-post-production strategy. They find that production strategy does not always benefit. Jung and Kouvelis [40] analyze the opportunities for co-operation between two firms based on the supply level that are rivals in the end-product market. Shi et al. [41] consider the encroaching manufacturer's internal organizational structure, where the different objectives of the manufacturer and e-commerce division alter the relationship in the decision process under the decentralized structure.

Subsequently, some researchers have shed light on the co-operation between platform firms and traditional markets. For example, Ozcan and Santos [42] care about the co-

opetitive between traditional mobile payments and financial platforms. Moreover, e-commerce platforms have emerged in many related modes recently. Li et al. [9] study the information sharing in an e-commerce platform containing an online marketplace, an upstream manufacturer and a reseller, then analyze four sharing scenarios and the equilibrium. Wu et al. [43] investigate a co-opetitive dual-channel supply chain wherein the supplier and e-commerce platform engage in a co-opetitive relationship with horizontal competition and vertical co-operation. Zhong et al. [44] investigated the incentives for sharing the forecast information with or without platform encroachment. For the co-operation between different platform firms, in a recent work, Cohen and Zhang [28] indicate two ride-sharing platform firms join in a profit-sharing contract by introducing a new joint service, and the authors show a well-designed profit-sharing contract. Wang et al. [45] focused on the platform-based co-opetitive supply chain whereby a flagship store operated by the manufacturer and a self-run store operated by the platform coexist in the platform market. Motivated by the above-related literature, we analyze the co-operation in online video platforms, especially broadcasting as the contract, which to the best of our knowledge, has not been studied before. In Table 1, we review the previous related studies and show how our work differs from theirs.

**Table 1.** Comparison with the related studies.

| Study | Two-Sided Market | | Online Video Platform | | Platform's Co-Opetitive Strategy |
|---|---|---|---|---|---|
| | Pricing Strategies | Multi-Homing | Video Industry | Streaming Content | |
| Fan et al. [15] | $\checkmark$ | - | - | - | - |
| Gal-Or et al. [18] | $\checkmark$ | - | - | - | - |
| Ambrus et al. [21] | $\checkmark$ | $\checkmark$ | $\checkmark$ | - | - |
| Athey et al. [46] | $\checkmark$ | $\checkmark$ | - | - | - |
| Li et al. [9] | $\checkmark$ | $\checkmark$ | - | - | $\checkmark$ |
| Xie et al. [19] | $\checkmark$ | $\checkmark$ | - | - | - |
| Amaldoss et al. [29] | $\checkmark$ | - | $\checkmark$ | $\checkmark$ | - |
| Cohen and Zhang [28] | $\checkmark$ | $\checkmark$ | - | - | $\checkmark$ |
| Our study | $\checkmark$ | $\checkmark$ | $\checkmark$ | $\checkmark$ | $\checkmark$ |

Our research contributes to the existing body of knowledge in the following two aspects. First, we shed light on the live streaming market, especially considering the duopoly platforms' co-opetitive strategy triggered by the broadcasting right of the content, which to the best of our knowledge, has not been analyzed under the cross-side network effects and multi-homing users through the analytic model. Second, we examine the multi-homing choices of users in a two-sided market framework and analyze platforms' pricing strategies, market share, and profit in three cases, including single-single, multi-single, and multi-multi subscribers and advertisers.

## 3. Problem Formulation

We consider a two-sided market consisting of three types of agents: co-opetitive platforms, subscribers, and advertisers. One platform is the primary rights-holding broadcaster (referred to as platform 1), such as China Central Television, which first obtains the broadcasting rights and sublicense agreement from the content creator (such as FIFA and OCOG) with the unit broadcasting rights cost $c_1$. In this case, platform 1 becomes the exclusive broadcaster if he does not authorize other platforms to get the re-broadcasting rights. However, the streaming content creators prefer a co-opetitive strategy to expand the market and often require the primary rights-holding broadcaster to resell the broadcasting rights. Thus, the secondary platform (referred to as platform 2) can get the re-broadcasting rights with the unit cost $c_2$, such as Migu online video platform. With the co-opetitive strategy, platforms offer homogeneous products (streaming content) to attract subscribers and provide commercial breaks for advertisers [6]. Generally, the unit broadcasting rights

costs $c_1$ and $c_2$ have a positive correlation with content diversity and quality, which are the main concerns of the subscribers. In addition to the broadcasting rights cost, the platforms also need to provide high-quality streaming services, such as a wider broadcasting range and shorter delay. As a result, the total cost of platforms, denoted as $c_i n_{Si}$, will be magnified by the number of subscribers $n_{Si}$.

In this two-sided market, the platforms constitute a co-opetitive structure, as shown in Figure 1, in which platform 1 serves as both the content provider (cooperator) and the market competitor to platform 2 [9].

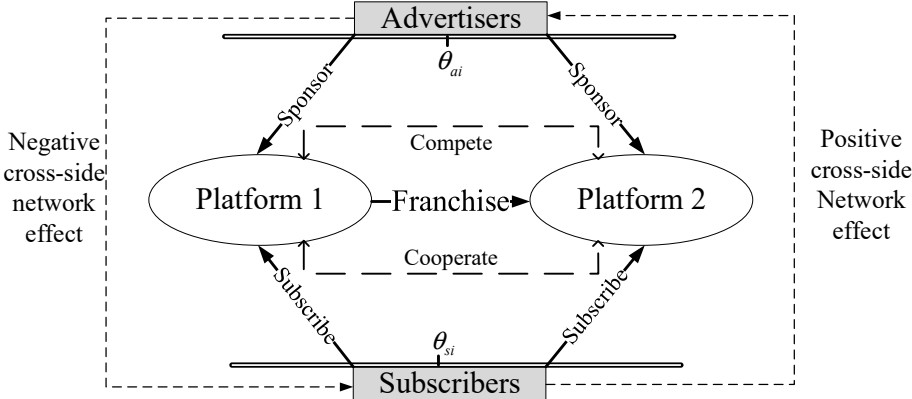

**Figure 1.** Platform's co-opetitive strategy in a two-sided market.

The decision sequence of the platforms is shown in Figure 2.

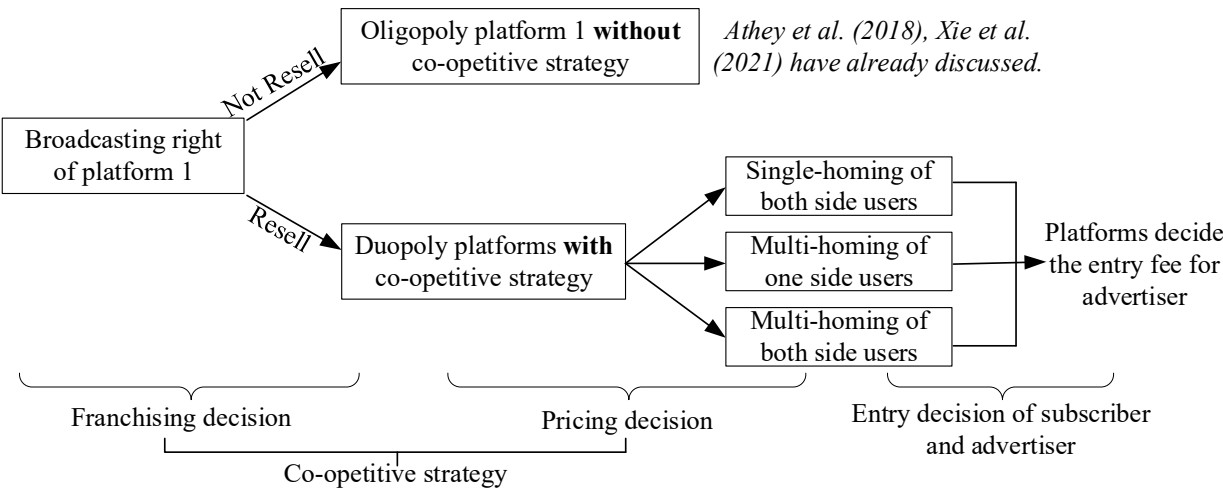

**Figure 2.** The sequence of events. [19,46].

We denote the subscribers as $s$ and the advertisers as $a$. Following Armstrong [4] and the basic assumption of the Hotelling model, users are uniformly distributed along a Hotelling line with a length of 1 and two platforms located at the terminal of the line. The utility of a subscriber choosing platform $i$ ($i = 1, 2$) is $v - \theta_{si} pr_s + \beta c_i - a_s n_{Ai}$, where $v$ is the reserved utility gained from the subscribed contents. Since each user has two options available in the market, they will incur the transportation costs, denoted by $pr_s$ and $pr_a$, respectively, for subscribers and advertisers. The transportation cost can reflect the user's preferences for the ideal platform. $\theta_{si}$ indicates the distance between the subscribers and platform $i$. Thus, the longer the distance, the stronger the user's willingness to choose the ideal platform and the higher the user's disutility of joining a platform away from the optimum. $c_i$ is the broadcasting rights cost of platform $i$; $\beta$ indicates the subscriber's preference for content quality; $n_{Ai}$ is the number of advertisers on the platform $i$; $a_s$

denotes the negative cross-side network effect of the subscriber for advertisements, the more advertisements, the larger disutility of subscribers. On the other hand, an advertiser needs to pay a certain amount of $ca_i$ to join the platform, whose utility on the platform $i$ is $v - \theta_{ai}pr_a + a_a n_{Si} - ca_i$, in which $n_{Si}$ denotes the number of subscribers on platform $i$; $a_a$ is the positive cross-side network effect of the subscribers. The summary of the notations and decision variables is provided in Table 2.

**Table 2.** Notations of parameters and variables.

| | |
|---|---|
| **Decision Variables** | |
| $ca_i$ | Entry fee charged by platform $i$ for advertiser. |
| **Parameters** | |
| $\theta_{ji}$ | Distance between platform and subscriber or advertiser, $j = s, a$. |
| $v$ | Reservation utility of subscriber |
| $pr_j$ | Subscriber or advertiser's transportation cost, $j = s, a$. |
| $\beta$ | Subscriber's preference for platform's content. |
| $c_i$ | Unit broadcasting rights cost of platform $i$ for content, $i = 1, 2$. |
| $a_s$ | Negative cross-side network effect of advertiser. |
| $a_a$ | Positive cross-side network effect of subscriber. |
| $n_{Si}$ | Number of the subscriber on platform $i = 1, 2, 12$. |
| $n_{Ai}$ | Number of the advertiser on platform $i = 1, 2, 12$. |
| $\pi_i$ | Profit function of platform $i$, $i = 1, 2$. |

### 3.1. Single-Homing Subscriber and Advertiser

To begin, we consider the case of single-homing subscribers and single-homing advertisers, in which the two-sided users only join one platform. According to the Hotelling model, two distinct indifference points can be used to divide the market demand of subscribers and advertisers on Platforms 1 and 2, respectively. We denote $\theta_j = \theta_{j1} = \theta_{j2}$. The subscriber and advertiser will choose one platform which can bring higher utility. Thus, the utility functions of single-homing subscribers on Platforms 1 and 2 are respectively as,

$$u_{S1} = v - pr_s\theta_s + \beta c_1 - a_s n_{A1} \tag{1}$$

$$u_{S2} = v - pr_s(1 - \theta_s) + \beta c_2 - a_s n_{A2} \tag{2}$$

Based on the heterogeneity of subscriber $\theta_s$, we obtain the equilibrium number of subscribers on Platforms 1 and 2, as $n_{S1}$ and $n_{S2}$, $n_{S1} = 1 - n_{S2} = \frac{\beta(c_1-c_2)-a_s(n_{A1}-n_{A2})+pr_s}{2pr_s}$. Similarly, the utility functions of single-homing advertisers on Platforms 1 and 2 are,

$$u_{A1} = v - \theta_a pr_a + a_a n_{S1} - ca_1 \tag{3}$$

$$u_{A2} = v - (1 - \theta_a)pr_a + a_a n_{S2} - ca_2 \tag{4}$$

From the rationale of the Hotelling model, we can find the location of the indifferent buyer and seller from the conditions of $u_{S1} = u_{S2}$ and $u_{A1} = u_{A2}$, respectively. Thus, the number of subscribers and advertisers on Platforms 1 and 2 can be obtained as follows:

$$n_{S1} = 1 - n_{S2} = \frac{1}{2} + \frac{a_s(ca_1-ca_2)+pr_a\beta(c_1-c_2)}{2(a_a a_s + pr_a pr_s)}.$$

$$n_{A1} = 1 - n_{A2} = \frac{1}{2} + \frac{a_a\beta(c_1-c_2)-pr_s(ca_1-ca_2)}{2(a_a a_s + pr_a pr_s)}. \tag{5}$$

With the co-opetitive strategy, platform 1 obtains the broadcasting right from the content creator with unit broadcasting rights cost $c_1$ and then resells it to platform 2. As to the online video platform, the demand for content will be large-scale and more heterogenous with the increasing number of subscribers, which requires the platform to make more efforts to improve the content quality, including stable internet connection, high-level service, and attractive content. Thus, the total cost of broadcasting is proportional to

the number of subscribers on the platform. The objective profit functions of Platforms 1 and 2 are

$$\max_{ca_1} \pi_1 = n_{A1}ca_1 + c_2 n_{S2} - c_1 n_{S1}.$$
$$\max_{ca_2} \pi_2 = n_{A2}ca_2 - c_2 n_{S2}.$$

(6)

Substituting Equations (5) into (6), we can calculate the first and second-order partial derivatives of profit functions with respect to $ca_1$ and $ca_2$, respectively. According to the first-order conditions of both Platforms 1 and 2, we have the optimal pricing decisions. Then, substituting the optimal results into Equations (5) and (6), the equilibrium market demand and profits of platforms can be obtained as Proposition 1.

**Proposition 1.** *In the case of single-homing users and advertisers in a two-sided market, the equilibrium pricing decisions of platforms are* $ca_1{}^{*s} = pr_a + \frac{3a_s a_a + a_a \beta(c_1 - c_2) - a_s(2c_1 + 3c_2)}{3pr_s}$, $ca_2{}^{*s} = pr_a + \frac{3a_s a_a - a_a \beta(c_1 - c_2) - a_s(c_1 + 3c_2)}{3pr_s}$. *The numbers of subscribers on platform 1 and 2 are* $n_{S1}{}^{*s} = \frac{1}{2} + \frac{\beta(c_1 - c_2)(3pr_a pr_s + 2a_a a_s) - a_s^2 c_1}{6pr_s(a_a a_s + pr_a pr_s)} = 1 - n_{S2}{}^{*s}$, *the numbers of advertisers on platforms are* $n_{A1}{}^{*s} = \frac{1}{2} + \frac{a_s c_1 + \beta a_a(c_1 - c_2)}{6(a_a a_s + pr_a pr_s)} = 1 - n_{A2}{}^{*s}$.

All the proofs are shown in Appendix A.

Proposition 1 investigates the platform's optimal pricing decisions when both the subscriber and the advertiser are single-homing. In equilibrium, the duopoly platforms divide the subscriber and advertiser market in a manner affected by the scale of network effect, subscriber preference, and broadcasting rights cost. Because all the users are single-homing, they will join one platform, which can provide higher utility. The result shows that the number of subscribers and the advertiser's entry fee of platform 1 is positively correlated with the subscriber's preference for content when $c_1 > c_2$; otherwise, it negatively correlated with the preference. In addition, by taking the derivative of equilibrium pricing to the key parameters, we have the following Corollary 1.

**Corollary 1.** *In the case of single-homing and advertisers in a duopoly market, the results of the sensitivity analysis reveal:*

(1) $\frac{\partial ca_1{}^{*s}}{\partial c_2} < 0$, $\frac{\partial ca_2{}^{*s}}{\partial c_1} < 0$, $\frac{\partial n_{S1}{}^{*s}}{\partial c_2} = -\frac{\partial n_{S2}{}^{*s}}{\partial c_2} < 0$, $\frac{\partial n_{A1}{}^{*s}}{\partial c_1} > 0$, $\frac{\partial n_{A1}{}^{*s}}{\partial c_2} < 0$.

(2) *When* $c_1 > c_2$, $\frac{\partial ca_1{}^{*s}}{\partial a_a} > 0$, $\frac{\partial ca_1{}^{*s}}{\partial \beta} > 0$, $\frac{\partial ca_2{}^{*s}}{\partial \beta} < 0$, $\frac{\partial n_{S1}{}^{*s}}{\partial \beta} = -\frac{\partial n_{S2}{}^{*s}}{\partial \beta} > 0$, $\frac{\partial n_{S1}{}^{*s}}{\partial a_s} = -\frac{\partial n_{S2}{}^{*s}}{\partial a_s} < 0$, $\frac{\partial n_{A1}{}^{*s}}{\partial \beta} > 0$.

Corollary 1 indicates that the higher the re-broadcasting rights cost of platform 2, the lower the optimal advertiser's entry fee of platform 1, and vice versa. This implies that when a platform pays a higher cost to enrich its streaming content, it will transfer the cost to advertisers to some extent, but at the same time, it will attract more consumers to improve the effectiveness of advertising. Furthermore, the number of subscribers on platform 1 decreases with the increase in re-broadcasting rights cost of platform 2 since the two platforms compete based on content richness in the market. When the primary platform invests more to obtain broadcasting rights, the number of advertisers on platform 1 will increase. However, as the secondary platform gains more broadcasting rights, this number will decline. This is because the number of advertisers is positively affected by the number of subscribers joining the platform.

When the broadcasting rights cost of platform 1 is higher than that of platform 2, the equilibrium result indicates that the entry fee charged by platform 1 for advertiser increases with the positive network effect of the subscriber and the subscriber's preference for the platform content; In contrast, the entry fee charged by platform 2 for advertiser decreases with the subscriber's preference for the platform content. Since when the subscribers in the market pay more attention to the richness of platform content, the primary platform

will gain more profits and a larger market share in the competition. Moreover, the number of subscribers on platform 1 increases with the subscriber's preference for the platform content, while the number of subscribers on platform 2 decreases with the preference. The result implies that the primary broadcasting platform can attract high-preference consumers to subscribe and gain a bigger share of the market.

### 3.2. Multi-Homing Subscriber and Single-Homing Advertiser

Second, we consider the case of partially multi-homing subscriber and single-homing advertiser, in which the subscribers have three options: subscribe to platform 1 ($u_{S1}$), platform 2 ($u_{S2}$), or simultaneously Platforms 1 and 2 ($u_{S12}$). Differing from single-homing, the multi-homing subscriber has more choices and services from two platforms but meanwhile suffers from both sides' advertisers. Figure 3 presents the partially multi-homing subscriber's structure with the co-opetitive strategy. In this case, there exists one indifference point on the advertiser side ($\theta_a$) and two on the subscriber side ($\theta_{s1}$ and $\theta_{s2}$). All the subscribers are divided into three segments on the Hotelling line. The left part $(0, \theta_{s1})$ represents the subscribers who are indifferent between single-homing on platform 1 and multi-homing on both platforms. The middle part $(\theta_{s1}, \theta_{s2})$ represents the multi-homing subscribers. The right part $(\theta_{s2}, 1)$ represents the indifference between multi-homing and single-homing on platform 2. Thus, $n_{A1} + n_{A2} = 1$ and $n_{S1} + n_{S2} + n_{S12} = 1$.

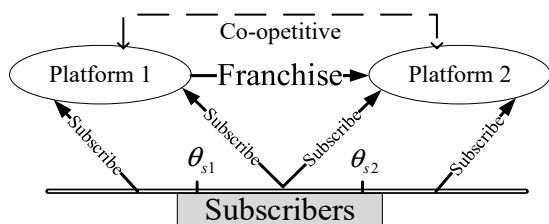

**Figure 3.** The case of partially multi-homing subscriber and single-homing advertiser.

The utility functions of subscribers in the multi-homing case are,

$$u_{S1} = v - pr_s\theta_{s1} + \beta c_1 - a_s n_{A1} \tag{7}$$

$$u_{S2} = v - pr_s(1 - \theta_{s2}) + \beta c_2 - a_s n_{A2} \tag{8}$$

$$u_{S12} = v - pr_s(\theta_{s2} - \theta_{s1}) + \beta(c_1 + c_2) - a_s(n_{A1} + n_{A2}) \tag{9}$$

Similarly, the utility functions of single-homing advertisers on Platforms 1 and 2 are,

$$u_{A1} = v - \theta_a pr_a + a_a(n_{S1} + n_{S12}) - ca_1 \tag{10}$$

$$u_{A2} = v - (1 - \theta_a)pr_a + a_a(n_{S2} + n_{S12}) - ca_2 \tag{11}$$

Thus, the number of subscribers $n_{S1}$, $n_{S2}$, $n_{S12}$ and single-homing advertisers on Platforms 1 and 2 can be obtained as follows, where $n_{S12} = \theta_{s2} - \theta_{s1}$:

$$
\begin{aligned}
n_{S1} &= 1 + \frac{a_s - \beta(c_1 + c_2)}{2pr_s} + \frac{a_s(ca_1 - ca_2) + \beta(c_1 - c_2)pr_a}{2(a_a a_s + pr_a pr_s)} \\[4pt]
n_{S2} &= 1 + \frac{a_s - \beta(c_1 + c_2)}{2pr_s} - \frac{a_s(ca_1 - ca_2) + \beta(c_1 - c_2)pr_a}{2(a_a a_s + pr_a pr_s)} \\[4pt]
n_{S12} &= \frac{\beta(c_1 + c_2) - a_s}{pr_s} - 1 \\[4pt]
n_{A1} &= 1 - n_{A2} = \frac{1}{2} + \frac{a_a\beta(c_1 - c_2) - (ca_1 - ca_2)pr_s}{2(a_a a_s + pr_a pr_s)}
\end{aligned}
\tag{12}
$$

The profit functions of platforms 1 and 2 are,

$$\max_{ca_1} \pi_1 = n_{A1} ca_1 + c_2(n_{S2} + n_{S12}) - c_1(n_{S1} + n_{S12})$$
$$\max_{ca_2} \pi_2 = n_{A2} ca_2 - c_2(n_{S2} + n_{S12})$$

(13)

Taking Equation (12) into (13), we can derive the optimal pricing decisions. Then, substituting the optimal results into Equations (12) and (13), the equilibrium market demand and profits of platforms can be obtained as Proposition 2.

**Proposition 2.** *In the case of partially multi-homing subscribers and single-homing advertisers, the equilibrium pricing decisions for advertisers are $ca_1^{*DS} = ca_1^{*s}$, $ca_2^{*DS} = ca_2^{*s}$. The number of multi-homing subscribers on platform 1 is $n_{S1}^{*DS} = 1 + \frac{-a_s^2 c_1 + 3(a_s - 2\beta c_2)pr_a pr_s + a_a a_s[3a_s - \beta(c_1 + 5c_2)]}{6pr_s(a_a a_s + pr_a pr_s)}$, satisfying $n_{S1}^{*DS} < n_{S1}^{*s}$, $n_{S2}^{*DS} = n_{S2}^{*s}$, $n_{S12}^{*DS} = \frac{\beta(c_1 + c_2) - a_s - pr_s}{pr_s}$. The number of advertisers satisfies $n_{A1}^{*DS} = 1 - n_{A2}^{*DS} = n_{A1}^{*s}$. The profits of platform $\pi_1^{*DS} < \pi_1^{*s}$ and $\pi_2^{*DS} < \pi_2^{*s}$.*

Proposition 2 indicates when the subscribers are partially multi-homing and advertisers are single-homing, the optimal pricing decisions of platforms exist, and the entry fee for advertisers is unchanged compared with the case of single-homing subscribers. The equilibriums show that the number of subscribers of platform 1 changes when the subscribers switch to multi-homing, but the number of advertisers does not. Moreover, the profits of two platforms both decrease with the multi-homing subscribers. By taking the derivative of equilibrium pricing to the key factors, we have the following Corollary 2.

**Corollary 2.** *In the case of partially multi-homing subscribers and single-homing advertisers, the results of the sensitivity analysis reveal:*

(1) $\frac{\partial n_{S12}^{*DS}}{\partial a_s} < 0$, $\frac{\partial n_{S12}^{*DS}}{\partial \beta} > 0$, $\frac{\partial n_{S12}^{*DS}}{\partial c_1} = \frac{\partial n_{S12}^{*DS}}{\partial c_2} > 0$.

(2) $\frac{\partial n_{S1}^{*DS}}{\partial \beta} < 0$, $\frac{\partial n_{S1}^{*DS}}{\partial c_1} < 0$, $\frac{\partial n_{S1}^{*DS}}{\partial c_2} < 0$. *When* $c_1 > c_2$, $\frac{\partial n_{S1}^{*DS}}{\partial pr_a} > 0$.

Corollary 2 indicates that, in the case of multi-homing subscribers and single-homing advertisers, a stronger negative network effect of advertisers results in a smaller number of multi-homing subscribers on both platforms, while greater subscriber content preference and higher broadcasting rights cost of platform result in a larger number of multi-homing subscribers. Furthermore, greater subscriber preference and higher broadcasting rights cost will both result in a smaller number of single-homing subscribers on platform 1. The results suggest that multi-homing subscribers are more susceptible to the negative network effect of the advertiser. On the one hand, when subscribing the live content from two platforms at the same time, the degree of interference from advertising is larger, which leads to a reduction in the number of live content subscribed from two platforms.

On the other hand, multi-homing subscribers will be positively affected by the richness of content and the degree of content preference. Thus, platforms could attract consumers by obtaining more content with a higher broadcasting rights cost or reducing the number of advertisers. However, to increase the number of single-homing subscribers, primary platform 1 prefers to lessen the broadcasting content. Furthermore, with the condition that the broadcasting rights cost of platform 1 is greater than that of platform 2, the number of single-homing subscribers on platform 1 increases with the single-homing advertiser's transportation cost.

### 3.3. Multi-Homing Subscriber and Advertiser

Single-homing advertisers are those who exclusively use the platform for advertising their products or services, whereas multi-homing advertisers also advertise on other platforms. In this section, we consider the case of both sides partially multi-homing, in which

the subscribers and advertisers have three options: join Platform 1, Platform 2, or Platforms 1 and 2. Thus, the utility functions of subscribers are,

$$u_{S1} = v - pr_s\theta_{s1} + \beta c_1 - a_s(n_{A1} + n_{A12}) \tag{14}$$

$$u_{S2} = v - pr_s(1 - \theta_{s2}) + \beta c_2 - a_s(n_{A2} + n_{A12}) \tag{15}$$

$$u_{S12} = v - pr_s(\theta_{s2} - \theta_{s1}) + \beta(c_1 + c_2) - a_s \tag{16}$$

Thus, the utility functions of advertisers are,

$$u_{A1} = v - \theta_{a1}pr_a + a_a(n_{S1} + n_{S12}) - ca_1 \tag{17}$$

$$u_{A2} = v - (1 - \theta_{a2})pr_a + a_a(n_{S2} + n_{S12}) - ca_2 \tag{18}$$

$$u_{A12} = v - pr_a(\theta_{a2} - \theta_{a1}) + a_a - ca_1 - ca_2 \tag{19}$$

where $n_{A12} = \theta_{a2} - \theta_{a1}$, the numbers of advertisers on Platforms 1 and 2 are,

$$n_{S1} = \frac{a_s(ca_1 + pr_a) - pr_a(\beta c_2 - pr_s)}{a_a a_s + pr_a pr_s}, \ n_{S2} = \frac{a_s(ca_2 + pr_a) - pr_a(\beta c_1 - pr_s)}{a_a a_s + pr_a pr_s}.$$

$$n_{S12} = \frac{a_a a_s - pr_a pr_s - a_s(ca_1 + ca_2 + 2pr_a) + \beta pr_a(c_1 + c_2)}{a_a a_s + pr_a pr_s}$$

$$n_{A1} = \frac{\beta a_a c_1 - (a_a - ca_2 - pr_a)pr_s}{a_a a_s + pr_a pr_s}, \ n_{A2} = \frac{\beta a_a c_2 - (a_a - ca_1 - pr_a)pr_s}{a_a a_s + pr_a pr_s}. \tag{20}$$

$$n_{A12} = \frac{a_a a_s - pr_a pr_s - a_a[\beta(c_1 + c_2) - 2pr_s] - (ca_1 + ca_2)pr_s}{a_a a_s + pr_a pr_s}.$$

The profit functions of Platforms 1 and 2 are,

$$\begin{aligned}\max_{ca_1} \pi_1 &= (n_{A1} + n_{A12})ca_1 + c_2(n_{S2} + n_{S12}) - c_1(n_{S1} + n_{S12}) \\ \max_{ca_2} \pi_2 &= (n_{A2} + n_{A12})ca_2 - c_2(n_{S2} + n_{S12})\end{aligned} \tag{21}$$

Taking Equations (21) into (20), we can derive the optimal pricing decisions. Then, substituting the optimal results into Equations (21) and (20), the equilibrium market demand and profits of platforms can be obtained as Proposition 3.

**Proposition 3.** *In the case of partially multi-homing subscribers and advertisers, the equilibrium pricing decisions for advertisers are* $ca_1{}^{*DD} = \frac{a_a(a_s - \beta c_2 + pr_s) - a_s c_2}{2pr_s}$, $ca_2{}^{*DD} = \frac{a_a(a_s - \beta c_1 + pr_s)}{2pr_s}$. *The number of subscribers on platform 1 is* $n_{S1}{}^{*DD} = \frac{pr_a(a_s - \beta c_2 + pr_s)}{a_a a_s + pr_a pr_s} + \frac{a_s a_a(a_s - \beta c_2 + pr_s) - a_s^2 c_2}{2pr_s(a_a a_s + pr_a pr_s)}$, $n_{S2}{}^{*DD} = \frac{(a_s - \beta c_1 + pr_s)(a_a a_s + 2pr_a pr_s)}{2pr_s(a_a a_s + pr_a pr_s)}$, $n_{S12}{}^{*DD} = \frac{a_s^2 c_2 + [\beta(c_1 + c_2) - 2a_s](a_a a_s + 2pr_a pr_s) - 2pr_a pr_s^2}{2pr_s(a_a a_s + pr_a pr_s)}$, *the number of advertisers on platform 1 is* $n_{A1}{}^{*DD} = \frac{a_a(a_s + \beta c_1 - pr_s) + 2pr_a pr_s}{2(a_a a_s + pr_a pr_s)}$, $n_{A2}{}^{*DD} = \frac{2pr_a pr_s - a_s c_2 + a_a(a_s + \beta c_2 - pr_s)}{2(a_a a_s + pr_a pr_s)}$, $n_{A12}{}^{*DD} = \frac{a_s c_2 - a_a\beta(c_1 + c_2) + 2a_a pr_s - 2pr_a pr_s}{2(a_a a_s + pr_a pr_s)}$.

Proposition 3 indicates that when the subscribers and advertisers are partially multi-homing, the platform's optimal pricing decisions, number of subscribers and advertisers, and profit of platforms exist. In equilibrium, the duopoly platforms divide the subscriber and advertiser market in a manner affected by the scale of network effects, preferences, and broadcasting rights costs. By taking the derivative of equilibrium pricing to the parameters, we have the following Corollary 3.

**Corollary 3.** *In the case of partially multi-homing subscribers and advertisers, the results of the sensitivity analysis reveal:*

(1) $\frac{\partial ca_i{}^{*DD}}{\partial \beta} < 0$, $\frac{\partial ca_i{}^{*DD}}{\partial c_{-i}} < 0$, $\frac{\partial ca_2{}^{*DD}}{\partial a_s} > 0$.

(2) $\frac{\partial n_{Si}{}^{*DD}}{\partial \beta} < 0$, $\frac{\partial n_{Si}{}^{*DD}}{\partial c_{-i}} < 0$, $\frac{\partial n_{S12}{}^{*DD}}{\partial \beta} > 0$, $\frac{\partial n_{S12}{}^{*DD}}{\partial c_1} > 0$, $\frac{\partial n_{S12}{}^{*DD}}{\partial c_2} > 0$.

(3)  $\frac{\partial n_{Ai}{}^{*DD}}{\partial \beta} > 0$, $\frac{\partial n_{A1}{}^{*DD}}{\partial c_1} > 0$. $\frac{\partial n_{A12}{}^{*DD}}{\partial \beta} < 0$, $\frac{\partial n_{A12}{}^{*DD}}{\partial c_1} < 0$.

Corollary 3 indicates that when the subscribers and advertisers are partially multi-homing, the entry fee charged by the platform *i* for the advertiser and the number of single-homing subscribers on the platform *i* decrease with the growth of subscriber's preference and the broadcasting rights cost of platforms $-i$. However, the number of multi-homing subscribers on platforms always increases with the growth of subscribers' preferences and broadcasting rights costs. In this case, the number of single-homing subscribers on one platform will decrease when the competing platform gets more content, but the number of multi-homing subscribers will increase with the broadcasting rights cost. Therefore, the platforms should pay more emphasis on increasing the content quality and diversity and then make efforts to improve the broadcasting service level when single-homing subscribers switch to multi-homing.

As to the advertisers, the number of single-homing advertisers on platform *i* increases, but the number of multi-homing advertisers decreases with the growth of subscriber preference. Moreover, the impacts of broadcasting rights cost on the number of multi-homing and single-homing advertisers are opposite. This result indicates that to attract more advertisers, the primary broadcasting platform can lessen content diversity with a lower cost when more single-homing advertisers switch to multi-homing.

## 4. Comparison among Cases

For a horizontal comparison, certain simplifications are required because our duopoly platform model with a two-sided market and co-opetitive strategy involves a range of factors with constraints that varies from case to case. In this section, we equalize the broadcasting rights costs of Platforms 1 and 2, then set the advertisers and subscribers to have the same transportation cost, which is $pr_a = pr_s = pr$ and $c_1 = c_2 = c$. We can obtain the equilibrium pricing decisions, market share, and profits under each case, as shown in Table 3.

**Table 3.** The comparison among different cases.

| | *S* | *DS* | *DD* |
|---|---|---|---|
| $ca_1$ | $pr - \frac{(5c-3a_a)a_s}{3pr}$. | $pr - \frac{(5c-3a_a)a_s}{3pr}$. | $\frac{a_a(pr-c\beta+a_s)-ca_s}{2pr}$. |
| $ca_2$ | $pr - \frac{(4c-3a_a)a_s}{3pr}$. | $pr - \frac{(4c-3a_a)a_s}{3pr}$. | $\frac{a_a(pr-c\beta+a_s)}{2pr}$. |
| $n_{S1}$ | $\frac{3pr^3+3pra_aa_s-ca_s^2}{6(pr^3+pra_aa_s)}$. | $1 - \frac{c\beta}{pr} + \frac{3pr^2a_s-a_s^2(c-3a_a)}{6(pr^3+pra_aa_s)}$. | $\frac{1}{2} - \frac{c\beta}{2pr} + \frac{2a_spr^2-a_s^2(c-a_a)}{2(pr^3+pra_aa_s)}$. |
| $n_{S2}$ | $\frac{3pr^3+3pra_aa_s+ca_s^2}{6(pr^3+pra_aa_s)}$. | $1 - \frac{c\beta}{pr} + \frac{3pr^2a_s+a_s^2(c+3a_a)}{6(pr^3+pra_aa_s)}$. | $\frac{(pr-c\beta+a_s)(2pr^2+a_aa_s)}{2(pr^3+pra_aa_s)}$. |
| $n_{S12}$ | / | $-\frac{pr-2c\beta+a_s}{pr}$. | $\frac{c\beta(2pr^2+a_aa_s)-pr^2(2a_s+pr)}{pr(a_aa_s+pr^2)} + \frac{(c-2a_a)a_s^2}{2pr(a_aa_s+pr^2)}$. |
| $n_{A1}$ | $\frac{1}{2} + \frac{ca_s}{6(pr^2+a_aa_s)}$. | $\frac{1}{2} + \frac{ca_s}{6(pr^2+a_aa_s)}$. | $\frac{2pr^2+a_a(c\beta+a_s-pr)}{2(pr^2+a_aa_s)}$. |
| $n_{A2}$ | $\frac{1}{2} - \frac{ca_s}{6(pr^2+a_aa_s)}$. | $\frac{1}{2} - \frac{ca_s}{6(pr^2+a_aa_s)}$. | $\frac{2pr^2-ca_s+a_a(c\beta+a_s-pr)}{2(pr^2+a_aa_s)}$. |
| $n_{A12}$ | / | / | $\frac{-2pr^2+2(pr-c\beta)a_a+ca_s}{2(pr^2+a_aa_s)}$. |
| $\pi_1$ | $\pi_1{}^{*s} = \frac{pr}{2} + \frac{(3a_a-4c)a_s}{6pr} + \frac{c^2a_s^2}{18pr(pr^2+a_aa_s)}$, | $\pi_1{}^{*DS} = \frac{pr^2+a_aa_s}{2pr} - \frac{2a_sc}{3pr} + \frac{c^2a_s^2}{18pr(pr^2+a_aa_s)}$, | $\pi_1{}^{*DD} = \frac{c^2a_s^2+a_a^2(pr-c\beta+a_s)^2}{4(pr^3+pra_aa_s)}$. |
| $\pi_2$ | $\pi_2{}^{*s} = \frac{pr-c}{2} + \frac{(3a_a-5c)a_s}{6pr} + \frac{c^2a_s^2}{18(pr^3+pra_aa_s)}$, | $\pi_2{}^{*DS} = \frac{pr^2-2c^2\beta+a_aa_s}{2pr} - \frac{a_sc}{3pr} + \frac{c^2a_s^2}{18pr(pr^2+a_aa_s)}$, | $\pi_2{}^{*DD} = \frac{a_a^2(pr-c\beta+a_s)^2}{4pr(pr^2+a_aa_s)} - \frac{ca_aa_s(pr+c\beta-a_s)+c(2cpr^2\beta-2pr^2a_s+ca_s^2)}{2pr(pr^2+a_aa_s)}$. |

**Conclusion 1.** *The optimal entry fee of platforms satisfies:* $ca_1{}^{*s} = ca_1{}^{*DS} < ca_2{}^{*s} = ca_2{}^{*DS}$, $ca_1{}^{*DD} < ca_2{}^{*DD}$. *When* $c < c\prime = \frac{3(2pr^2 - a_a pr + a_a a_s)}{7a_s - 3a_a\beta}$, $ca_1{}^{*s} > ca_1{}^{*DD}$; *When* $c < c'' = \frac{3(2pr^2 - a_a pr + a_a a_s)}{8a_s - 3a_a\beta}$, $ca_2{}^{*s} > ca_2{}^{*DD}$. $c'' < c\prime$.

In our model, platform 1 firstly obtains broadcasting right and then authorizes the re-broadcasting rights to platform 2. Conclusion 1 indicates that when $pr_a = pr_s = pr$ and $c_1 = c_2 = c$, platform 1 always charges a lower entry fee for advertisers than platform 2 in all three cases. The findings suggest that in a two-sided market with co-opetitive platforms, the advertisers should join the platform having prior broadcasting rights to lower the marginal cost of advertising expenditure. Furthermore, when the broadcasting rights cost is less than a threshold $c\prime$, the advertiser's entry fee charged by platform 1 will decrease in the case of multi-homing users than single-homing. When the broadcasting rights cost is less than a threshold $c''$, the entry fee charged by platform 2 gets lower in the case of multi-homing users. Accordingly, the advertisers are more willing to migrate to multi-homing if the broadcasting rights cost is lower than a certain threshold. Otherwise, they would switch to single-homing and join the primary broadcasting platform.

**Conclusion 2.** *The number of subscribers satisfies:* $n_{S2}{}^{*s} > n_{S1}{}^{*s} > n_{S1}{}^{*DS}$, $n_{S2}{}^{*s} > n_{S2}{}^{*DS} > n_{S1}{}^{*DS}$, $n_{S1}{}^{*s} < n_{S1}{}^{*DS} + n_{S12}{}^{*DS}$, $n_{S2}{}^{*s} < n_{S2}{}^{*DS} + n_{S12}{}^{*DS}$. *When* $c < \frac{2pr(pr - 2a_a)}{a_s - 2\beta a_a}$, $n_{S12}{}^{*DS} > n_{S12}{}^{*DD}$; *When* $c > c\prime = \frac{3pr^2(pr + 2a_s) + 3a_s^2 a_a}{6\beta pr^2 + a_s^2 + 3a_s a_a\beta}$, $n_{S1}{}^{*s} < n_{S1}{}^{*DD} + n_{S12}{}^{*DD}$; *When* $c > c'' = \frac{3pr^2(pr + 2a_s) + 3a_s^2 a_a}{6\beta pr^2 + 2a_s^2 + 3a_s a_a\beta}$, $n_{S2}{}^{*s} < n_{S2}{}^{*DD} + n_{S12}{}^{*DD}$. $c'' < c\prime$. *The number of advertisers satisfies:* $n_{A1}{}^{*s} = n_{A1}{}^{*DS} > n_{A2}{}^{*s} = n_{A2}{}^{*DS}$. *When* $c > \frac{3pr(pr - a_a)}{2a_s - 3a_a\beta}$, $n_{A1}{}^{*DS} < n_{A1}{}^{*DD} + n_{A12}{}^{*DD}$; *When* $c > \frac{3pr(pr - a_a)}{a_s - 3a_a\beta}$, $n_{A2}{}^{*DS} < n_{A2}{}^{*DD} + n_{A12}{}^{*DD}$.

Conclusion 2 shows that although there are more subscribers joining platform 2 than platform 1 when advertisers are single-homing, there are fewer advertisers joining platform 2. Because platform 1 has primary broadcasting rights, it can decrease the entry fee for advertisers to increase the number of advertisers and gain more revenue, despite the fact that the advertisers' negative network effect will result in a decline in subscriber numbers. Furthermore, the numbers of subscribers and advertisers are always greater in the case of partially multi-homing advertisers than single-homing only when the broadcasting rights cost is relatively higher than certain thresholds. As a result, when the broadcasting rights cost, that is, the content diversity and quality of the platform, have been improved, the platforms with a co-opetitive strategy prefer the advertisers to join more than one platform at the same time to broaden the consumer market.

**Conclusion 3.** *The optimal profit of the platform satisfies:* $\pi_1{}^{*s} = \pi_1{}^{*DS} > \pi_2{}^{*s} > \pi_2{}^{*DS}$; *When* $c > \frac{4pr^2 a_s - 2a_s a_a(pr - a_s)}{4pr^2\beta + 2a_s a_a\beta + 3a_s^2}$, $\pi_1{}^{*DD} > \pi_2{}^{*DD}$.

Conclusion 3 analyzes that when the advertisers are single-homing, the profit of platform 1 is always higher than that of platform 2. In the case of two sides multi-homing, the profit of platform 1 gets higher than that of platform 2 only when the broadcasting rights cost exceeds a certain threshold. This result is consistent with the intuition that the primary broadcasting platform can gain more profit than its competitor by vying for the users. We further compare the platforms' profits in three cases based on the numerical analysis in the next section.

## 5. Numerical Analysis

In this section, we conduct numerical analysis to verify the impacts of the network effects $a_a$, $a_s$, the re-broadcasting rights cost $c_2$, on the number of subscribers and advertisers and the profits of platforms in three cases. The simulation results of three scenarios are shown in Figures 4–7. First, we analyze the impacts of re-broadcasting rights cost on the profits of two platforms in different cases, as shown in Figure 4. To ensure the existence

of equilibriums in three cases, the values of parameters are set as $\beta = 1$, $c_1 = 0.5$, $pr_a = pr_s = 0.8$, $a_s = a_a = 0.7$.

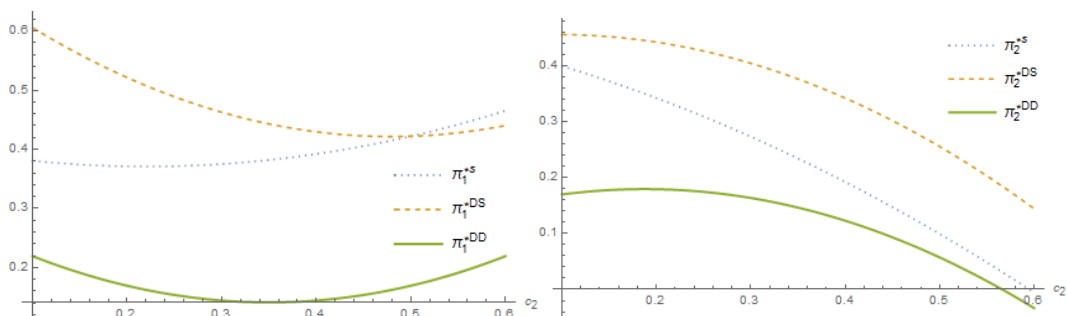

**Figure 4.** The impacts of re-broadcasting rights cost on profits in three cases.

Figure 4 indicates that with the growth of re-broadcasting rights cost, the optimal profit of platform 1 with multi-homing subscribers declines first, then increases. The profit from a single-homing subscriber, on the one hand, always rises in tandem with the re-broadcasting rights cost. Platform 1, with multi-homing subscribers and advertisers, on the other hand, has the lowest profit of the three scenarios. Interestingly, platform 1 prefers single-homing advertisers over multi-homing ones, which can bring higher profits. A higher re-broadcasting rights cost will increase the profit of platform 1 when both the advertisers and subscribers are single-homing. However, when the advertisers are single-homing, but the subscribers are multi-homing, platform 1 prefers a lower re-broadcasting rights cost, that is, authorizes fewer contents to platform 2.

The profit of platform 2 fluctuates in the exact opposite direction. Figure 4 demonstrates that the profit of platform 2 decreases as the re-broadcasting rights cost rises. Similarly, the profit of platform 2 with multi-homing subscribers and advertisers is the lowest among the three cases. In addition, the increase in re-broadcasting rights costs will reduce the profit margin of platform 2. When the advertisers switch to multi-homing, the profits of both platforms will significantly decrease due to the smaller number of subscribers.

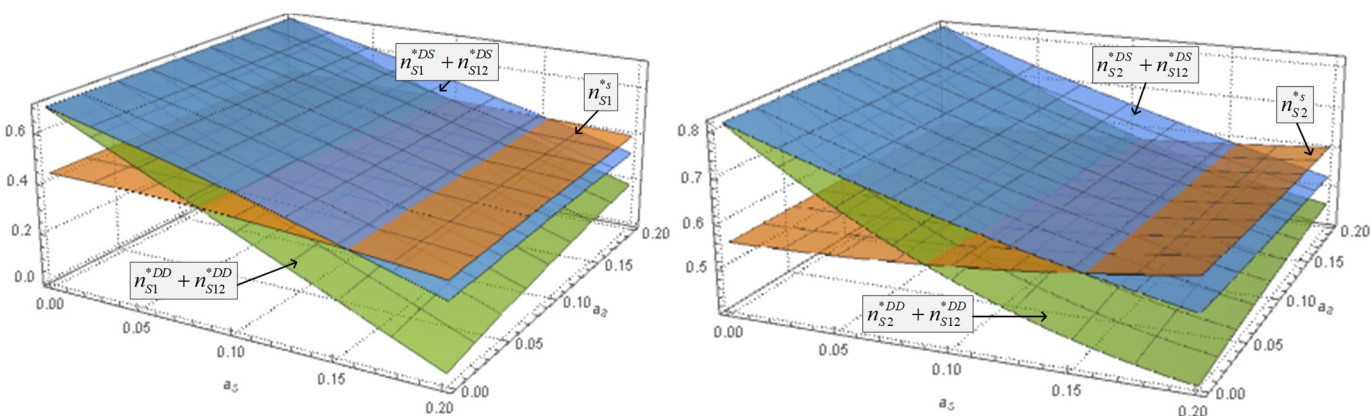

**Figure 5.** The impacts of network effect on the number of subscribers in three cases.

Figures 5 and 6 depict the equilibrium number of subscribers and advertisements with respect to the positive and negative network effects, respectively. To ensure the existence of equilibrium solutions in three cases, the values of the remaining parameters are set as $\beta = 0.7$, $c_1 = 0.3$, $c_2 = 0.35$, $pr_a = pr_s = 0.3$. Figure 5 further indicates that as the advertiser's negative network effect grows, the number of subscribers on platform 1 decreases. The result is consistent with Corollary 1 and Corollary 2. When the negative network effect is lower enough, the number of subscribers reaches the highest in the multi-single case, the second highest in the multi-multi case, and the least in the single-single

case. However, with the increase of negative network effect, the platform will gain the most subscribers in the single-single case, followed by the multi-single case, and finally, the multi-multi case. Therefore, as the disutility of advertising to subscribers increases, we suggest the platforms limit the entrance scale of the multi-homing advertisers.

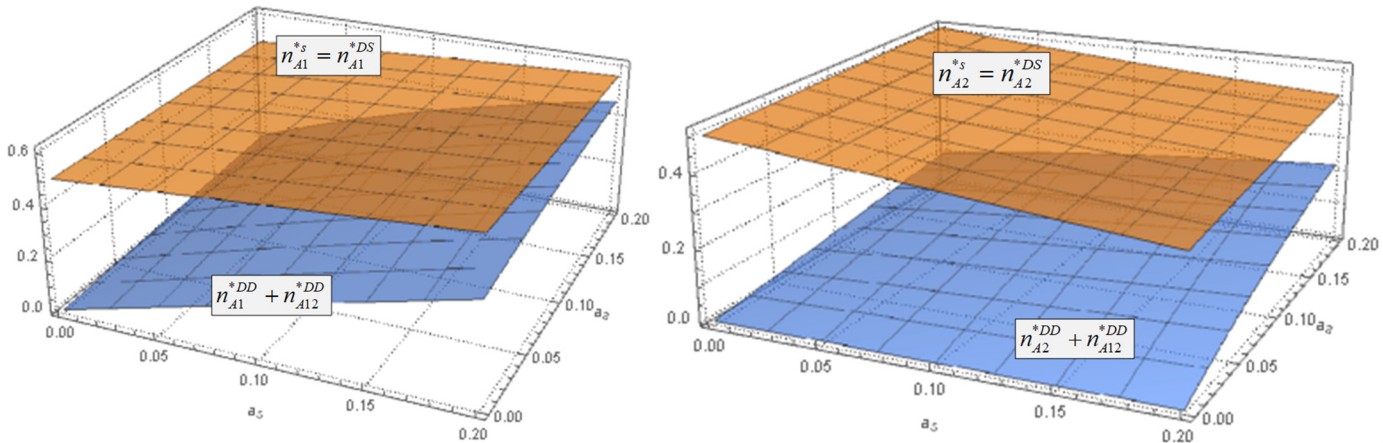

**Figure 6.** The impacts of network effect on the number of advertisers in three cases.

Figure 6 illustrates that when a platform has only single-homing advertisers, the total number of advertisers on the platform ($n_{Ai}{}^{*s} = n_{Ai}{}^{*DS}$) is higher compared to a scenario where there are multi-homing advertisers ($n_{Ai}{}^{*DD} + n_{A12}{}^{*DD}$). For the platforms that rely on advertising revenue, having fewer advertisers would result in lower income. Therefore, the profit of the platform will decrease when some of its advertisers switch to using multiple platforms instead of one. Furthermore, as the cross-side network effects increase, the numerical difference between the two scenarios reduces. This is because, as the positive cross-side network effect $a_a$ decreases, it becomes less beneficial for multi-homing advertisers to advertise on more than one platform, leading to a decline in the total number of advertisers. Based on the results, we suggest that platforms should focus on building strong cross-side network effects with multi-homing advertisers, which can lead to a larger overall pool of advertisers and higher revenue. Alternatively, if the cross-side network effects are relatively low, the platforms may be better off contracting with single-homing advertisers.

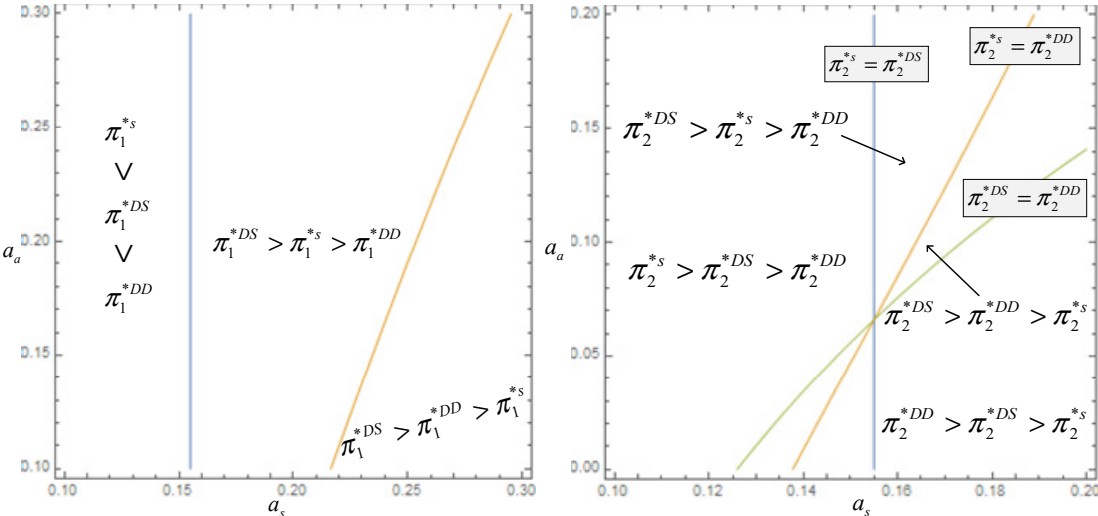

**Figure 7.** Comparisons of platforms' profits in three cases.

Figure 7 compares the optimal profits in different cases. The results indicate that when the negative network effect of the advertiser is less than the threshold, the profit of platform 1 is higher in the case of single-homing subscribers than multi-homing subscribers. The profit of Platform 1 in the multi-single case increases as the negative network effect rises, whereas the profit of Platform 1 in the single-homing subscriber's case decreases. Furthermore, when the negative network effect is higher than the threshold, the profit of platform 1 in the single-single case is the lowest. Thus, when the multi-homing subscribers are more sensitive to advertisements, the platforms should only allow the single-homing advertisers to join. When the subscriber's positive network effect is relatively higher, the profit of platform 2 in the case of single-single users gradually decreases as the negative network effect increases, while the profit of platform 2 in the case of multi-single users gradually increases. Therefore, the re-broadcasting platform prefers multi-homing subscribers and single-homing advertisers when the subscribers are more sensitive to commercials. Overall, when subscribers are multi-homing, but advertisers are single-homing, platforms will gain more profit in a co-opetitive situation.

## 6. Conclusions

As the live streaming industry develops, platform firms that implement a co-opetitive strategy become more profitable in a two-sided market with multi-homing users, but they also face the challenge of making the pricing strategy without losing the market share. Therefore, we focus on the co-opetitive strategy optimization of duopoly platforms and solve the pricing problems, utilize the extended Hotelling model to describe the user's behavior, and derive the equilibrium numbers of users on different platforms. Then, we analyze the equilibrium results of pricing, market share and platforms' profit based on the Bertrand duopoly model in three scenarios: single-single, multi-single, and multi-multi users. We find that advertisers only choose to be multi-homing with multi-homing subscribers when the broadcasting rights cost is relatively low. Otherwise, they prefer to join the primary broadcasting platform over the re-broadcasting platform when the subscribers are single-homing.

Furthermore, we find that the primary broadcasting platform always earns more than the re-broadcasting platform with single-homing advertisers. However, when more advertisers switch to multi-homing, the primary broadcasting platform can increase its profit if the broadcasting rights cost is relatively high. We also observe that when both both-side users are single-homing, the primary broadcasting platform prefers to authorize more content to the re-broadcasting platform than in the case of multi-homing subscribers. This implies that the platforms may lose profit when advertisers join both platforms simultaneously. Thus, we suggest that the platforms should focus on building strong cross-side network effects with multi-homing advertisers. Alternatively, they would be better off contracting with single-homing advertisers if the effects are relatively low.

There are some limitations to our study. For instance, we consider only two co-opetitive platforms in the market while ignoring the possibility of third-party competition. Besides, although our work sheds light on theoretical analysis, our findings can be empirically verified by a field study, which will contribute to the practice of enterprises. Based on the theoretical findings derived from this model, such as the optimal decisions and trends between variables, we will further investigate the co-opetitive strategy of the platforms through the empirical method, which will be of great managerial and academic value.

**Author Contributions:** Conceptualization, S.G.; methodology, J.L. and S.G.; software, S.G.; formal analysis, J.L.; resources, J.L. and X.L.; writing—original draft preparation, S.G.; writing—review and editing, J.L. and X.L.; visualization, X.L.; supervision, J.L.; funding acquisition, J.L. and S.G. All authors have read and agreed to the published version of the manuscript.

**Funding:** This research was funded by the National Social Science Foundation of China, grant number [22CGL020].

**Institutional Review Board Statement:** Not applicable.

**Informed Consent Statement:** Not applicable.

**Data Availability Statement:** This work is based on stylized economic models rather than empirical data. We use no data in this work.

**Conflicts of Interest:** The authors declare no conflict of interest.

## Appendix A

**Proof of Proposition 1.** The second order condition $\frac{\partial^2 \pi_1}{\partial ca_1{}^2} = \frac{\partial^2 \pi_2}{\partial ca_2{}^2} = -\frac{pr_s}{pr_a pr_s + a_a a_s} < 0$ grantee the strict concavity of profit functions with respect to pricing decisions, then we can drive the optimal pricing decisions based on FOC,

$$ca_1{}^{*s} = pr_a + \frac{3a_s a_a + a_a \beta(c_1 - c_2) - a_s(2c_1 + 3c_2)}{3pr_s}$$

$$ca_2{}^{*s} = pr_a + \frac{3a_s a_a - a_a \beta(c_1 - c_2) - a_s(c_1 + 3c_2)}{3pr_s}$$

Therefore, the equilibrium number of subscribers and advertisers on platforms is,

$$n_{S1}{}^{*s} = 1 - n_{S2}{}^{*} = \frac{1}{2} + \frac{\beta(c_1 - c_2)(3pr_a pr_s + 2a_a a_s) - a_s^2 c_1}{6pr_s(a_a a_s + pr_a pr_s)},$$

$$n_{A1}{}^{*s} = 1 - n_{A2}{}^{*} = \frac{1}{2} + \frac{a_s c_1 + \beta a_a(c_1 - c_2)}{6(a_a a_s + pr_a pr_s)}.$$

Taking the optimal decisions into profit functions, we can derive the equilibrium platform profits,

$$\pi_1{}^{*s} = \frac{[a_s c_1 + \beta a_a(c_1 - c_2)]^2}{18pr_s(a_a a_s + pr_a pr_s)} - \frac{c_1 - c_2 - pr_a}{2} + \frac{a_a[9a_s + 6\beta(c_1 - c_2)] - 3a_s(c_1 + 3c_2) - 9\beta(c_1^2 - c_2^2)}{18pr_s},$$

$$\pi_2{}^{*s} = \frac{[a_s c_1 + \beta a_a(c_1 - c_2)]^2}{18pr_s(a_a a_s + pr_a pr_s)} - \frac{c_1 - c_2 - pr_a}{2} + \frac{\beta(c_1 - c_2)(3c_2 - 2a_a) - a_s(2c_1 + 3c_2) + 3a_s a_a + 3(c_1 - 2c_2)pr_s}{6pr_s}.$$

$\square$

**Proof of Corollary 1.** Analyze the sensitivity of equilibrium results under the case of single-homing users, yield:

For the optimal pricing decision of platform 1, $ca_1{}^{*s}$, $\frac{\partial ca_1{}^{*s}}{\partial a_a} = \frac{3a_s + \beta c_1 - \beta c_2}{3pr_s}$, $\frac{\partial ca_1{}^{*s}}{\partial \beta} = \frac{a_a(c_1 - c_2)}{3pr_s}$, $\frac{\partial ca_1{}^{*s}}{\partial c_2} = -\frac{\beta a_a + 3a_s}{3pr_s}$ for the optimal pricing decision of platform 2, $\frac{\partial ca_2{}^{*s}}{\partial \beta} = -\frac{a_a(c_1 - c_2)}{3pr_s}$, $\frac{\partial ca_2{}^{*s}}{\partial c_1} = -\frac{\beta a_a + a_s}{3pr_s}$.

For the equilibrium number of subscriber, $\frac{\partial n_{S1}{}^{*s}}{\partial a_s} = -\frac{\partial n_{S2}{}^{*s}}{\partial a_s} = \frac{-a_a a_s^2 c_1 - (2a_s c_1 + \beta a_a(c_1 - c_2))pr_a pr_s}{6pr_s(a_a a_s + pr_a pr_s)^2}$, $\frac{\partial n_{S1}{}^{*s}}{\partial \beta} = -\frac{\partial n_{S2}{}^{*s}}{\partial \beta} = \frac{(c_1 - c_2)(2a_a a_s + 3pr_a pr_s)}{6pr_s(a_a a_s + pr_a pr_s)}$, $\frac{\partial n_{S1}{}^{*s}}{\partial c_2} = -\frac{\partial n_{S2}{}^{*s}}{\partial c_2} = -\frac{\beta}{6}\left(\frac{2}{pr_s} + \frac{pr_a}{a_a a_s + pr_a pr_s}\right)$. For the equilibrium number of advertiser, $\frac{\partial n_{A1}{}^{*s}}{\partial \beta} = -\frac{\partial n_{A2}{}^{*s}}{\partial \beta} = \frac{a_a(c_1 - c_2)}{6(a_a a_s + pr_a pr_s)}$, $\frac{\partial n_{A1}{}^{*s}}{\partial c_1} = -\frac{\partial n_{A2}{}^{*s}}{\partial c_1} = \frac{\beta a_a + a_s}{6a_a a_s + 6pr_a pr_s}$, $\frac{\partial n_{A1}{}^{*s}}{\partial c_2} = -\frac{\partial n_{A2}{}^{*s}}{\partial c_2} = -\frac{\beta a_a}{6(a_a a_s + pr_a pr_s)}$. $\square$

**Proof of Proposition 2.** According to $\frac{\partial^2 \pi_1}{\partial ca_1{}^2} = \frac{\partial^2 \pi_2}{\partial ca_2{}^2} = -\frac{pr_s}{pr_a pr_s + a_a a_s} < 0$ and FOC, we can get the optimal pricing decisions $ca_1{}^{*DS}$ and $ca_2{}^{*DS}$. Therefore, the equilibrium number of subscriber and advertiser are,

$$n_{S1}{}^{*DS} = 1 + \frac{-a_s^2 c_1 + 3(a_s - 2\beta c_2)pr_a pr_s + a_a a_s[3a_s - \beta(c_1 + 5c_2)]}{6pr_s(a_a a_s + pr_a pr_s)}$$

$$n_{S2}{}^{*DS} = 1 + \frac{a_s^2 c_1 + 3(a_s - 2\beta c_1)pr_a pr_s + a_a a_s[3a_s - \beta(5c_1 + c_2)]}{6pr_s(a_a a_s + pr_a pr_s)}$$

$$n_{S12}{}^{*DS} = \frac{\beta(c_1 + c_2) - a_s - pr_s}{pr_s}$$

$$n_{A1}{}^{*DS} = 1 - n_{A2}{}^{*DS} = \frac{1}{2} + \frac{a_s c_1 + \beta a_a(c_1 - c_2)}{6(a_a a_s + pr_a pr_s)}$$

$$\pi_1{}^{*DS} = \frac{a_a[3a_s + 2\beta(c_1 - c_2)] + 2a_s(c_1 - 3c_2)}{6pr_s} - \frac{\beta(c_1^2 - c_2^2)}{pr_s} + \frac{pr_a}{2} + \frac{[a_s c_1 + \beta a_a(c_1 - c_2)]^2}{18pr_s(a_a a_s + pr_a pr_s)}$$

$$\pi_2{}^{*DS} = \frac{a_a[3a_s - 2\beta(c_1 - c_2)] - 2(a_s c_1 + 3\beta c_2^2)}{6pr_s} + \frac{pr_a}{2} + \frac{[a_s c_1 + \beta a_a(c_1 - c_2)]^2}{18pr_s(a_a a_s + pr_a pr_s)}$$

To ensure $n_{S12}{}^{*DS} > 0$, the assumption is required $\beta(c_1 + c_2) > a_s + pr_s$. Then, compare the results with Proposition 1; the relationships satisfy: $n_{S1}{}^{*DS} - n_{S1}{}^{*s} = \frac{a_s - \beta(c_1 + c_2) + pr_s}{2pr_s} < 0$, $n_{S2}{}^{*DS} - n_{S2}{}^{*s} = 0$, $n_{A1}{}^{*DS} - n_{A1}{}^{*s} = 0$, $\pi_1{}^{*DS} - \pi_1{}^{*s} = \frac{(c_1 - c_2)(a_s - \beta(c_1 + c_2) + pr_s)}{2pr_s} < 0$, $\pi_2{}^{*DS} - \pi_2{}^{*s} = \frac{c_2(a_s - \beta(c_1 + c_2) + pr_s)}{2pr_s} < 0$. □

**Proof of Corollary 2.** Analyze the sensitivity of equilibrium results under the case of multi-homing subscriber and single-homing advertiser, yield:

$$\frac{\partial n_{S12}{}^{*DS}}{\partial a_s} = -\frac{1}{pr_s}, \frac{\partial n_{S12}{}^{*DS}}{\partial \beta} = \frac{c_1 + c_2}{pr_s}, \frac{\partial n_{S12}{}^{*DS}}{\partial c_1} = \frac{\partial n_{S12}{}^{*DS}}{\partial c_2} = \frac{\beta}{pr_s}.$$

$$\frac{\partial n_{S1}{}^{*DS}}{\partial pr_a} = \frac{a_s(a_s c_1 + \beta a_a(c_1 - c_2))}{6(a_a a_s + pr_a pr_s)^2}, \frac{\partial n_{S1}{}^{*DS}}{\partial \beta} = -\frac{a_a a_s(c_1 + 5c_2) + 6c_2 pr_a pr_s}{6pr_s(a_a a_s + pr_a pr_s)},$$

$$\frac{\partial n_{S1}{}^{*DS}}{\partial c_1} = -\frac{a_s(\beta a_a + a_s)}{6pr_s(a_a a_s + pr_a pr_s)}, \frac{\partial n_{S1}{}^{*DS}}{\partial c_2} = -\frac{\beta}{6}\left(\frac{5}{pr_s} + \frac{pr_a}{a_a a_s + pr_a pr_s}\right).$$

□

**Proof of Proposition 3.** According to $\frac{\partial^2 \pi_1}{\partial ca_1^2} = \frac{\partial^2 \pi_2}{\partial ca_2^2} = -\frac{2pr_s}{pr_a pr_s + a_a a_s} < 0$ and FOC, we can get the optimal pricing decisions $ca_1{}^{*DD}$ and $ca_2{}^{*DD}$. Therefore, the equilibrium number of subscriber and advertiser are,

$$n_{S1}{}^{*DD} = \frac{pr_a(a_s - \beta c_2 + pr_s)}{a_a a_s + pr_a pr_s} + \frac{a_s a_a(a_s - \beta c_2 + pr_s) - a_s^2 c_2}{2pr_s(a_a a_s + pr_a pr_s)}, n_{S2}{}^{*DD} = \frac{(a_s - \beta c_1 + pr_s)(a_a a_s + 2pr_a pr_s)}{2pr_s(a_a a_s + pr_a pr_s)}.$$

$$n_{S12}{}^{*DD} = \frac{a_s^2 c_2 + [\beta(c_1 + c_2) - 2a_s](a_a a_s + 2pr_a pr_s) - 2pr_a pr_s^2}{2pr_s(a_a a_s + pr_a pr_s)}.$$

$$n_{A1}{}^{*DD} = \frac{a_a(a_s + \beta c_1 - pr_s) + 2pr_a pr_s}{2(a_a a_s + pr_a pr_s)}, n_{A2}{}^{*DD} = \frac{2pr_a pr_s - a_s c_2 + a_a(a_s + \beta c_2 - pr_s)}{2(a_a a_s + pr_a pr_s)}.$$

$$n_{A12}{}^{*DD} = \frac{a_s c_2 - a_a \beta(c_1 + c_2) + 2a_a pr_s - 2pr_a pr_s}{2(a_a a_s + pr_a pr_s)}.$$

$$\pi_1{}^{*DD} = \frac{a_s^2 c_2^2 + 2(c_1 - c_2)[a_s - \beta(c_1 + c_2)](2pr_a pr_s + a_a a_s) + a_a^2(a_s - \beta c_2 + pr_s)^2}{4pr_s(a_a a_s + pr_a pr_s)} - \frac{a_a a_s(c_1 - c_2)}{2(a_a a_s + pr_a pr_s)}.$$

$$\pi_2{}^{*DD} = \frac{-2a_s^2 c_2^2 + 2c_2(a_s - \beta c_2)(a_a a_s + 2pr_a pr_s) + a_a^2(a_s - \beta c_1 + pr_s)^2}{4pr_s(a_a a_s + pr_a pr_s)} - \frac{a_a a_s c_2}{2(a_a a_s + pr_a pr_s)}.$$

□

**Proof of Corollary 3.** Analyze the sensitivity of equilibrium results under the case of multi-homing subscriber and advertiser, yield:

$$\frac{\partial ca_1{}^{*DD}}{\partial \beta} = -\frac{a_a c_2}{2pr_s}, \ \frac{\partial ca_1{}^{*DD}}{\partial c_2} = -\frac{\beta a_a + a_s}{2pr_s}, \ \frac{\partial ca_2{}^{*DD}}{\partial c_1} = -\frac{\beta a_a}{2pr_s}, \ \frac{\partial ca_2{}^{*DD}}{\partial \beta} = -\frac{a_a c_1}{2pr_s}, \ \frac{\partial ca_2{}^{*DD}}{\partial a_s} = \frac{a_a}{2pr_s}.$$

$$\frac{\partial n_{S1}{}^{*DD}}{\partial \beta} = -\frac{c_2}{2}\left(\frac{1}{pr_s} + \frac{pr_a}{a_a a_s + pr_a pr_s}\right), \ \frac{\partial n_{S1}{}^{*DD}}{\partial c_2} = -\frac{\beta a_a a_s + a_s^2 + 2\beta pr_a pr_s}{2a_a a_s pr_s + 2pr_a pr_s^2},$$

$$\frac{\partial n_{S2}{}^{*DD}}{\partial \beta} = -\frac{c_1}{2}\left(\frac{1}{pr_s} + \frac{pr_a}{a_a a_s + pr_a pr_s}\right), \ \frac{\partial n_{S2}{}^{*DD}}{\partial c_1} = -\frac{\beta}{2}\left(\frac{1}{pr_s} + \frac{pr_a}{a_a a_s + pr_a pr_s}\right).$$

$$\frac{\partial n_{S12}{}^{*DD}}{\partial \beta} = \frac{c_1 + c_2}{2}\left(\frac{1}{pr_s} + \frac{pr_a}{a_a a_s + pr_a pr_s}\right), \ \frac{\partial n_{S12}{}^{*DD}}{\partial c_1} = \frac{\beta}{2}\left(\frac{1}{pr_s} + \frac{pr_a}{a_a a_s + pr_a pr_s}\right),$$

$$\frac{\partial n_{S12}{}^{*DD}}{\partial c_2} = \frac{\beta a_a a_s + a_s^2 + 2\beta pr_a pr_s}{2a_a a_s pr_s + 2pr_a pr_s^2}.$$

$$\frac{\partial n_{A1}{}^{*DD}}{\partial \beta} = \frac{a_a c_1}{2a_a a_s + 2pr_a pr_s}, \ \frac{\partial n_{A1}{}^{*DD}}{\partial c_1} = \frac{\beta a_a}{2a_a a_s + 2pr_a pr_s}, \ \frac{\partial n_{A2}{}^{*DD}}{\partial \beta} = \frac{a_a c_2}{2a_a a_s + 2pr_a pr_s}.$$

$$\frac{\partial n_{A12}{}^{*DD}}{\partial \beta} = -\frac{a_a(c_1 + c_2)}{2(a_a a_s + pr_a pr_s)}, \ \frac{\partial n_{A12}{}^{*DD}}{\partial c_1} = -\frac{\beta a_a}{2(a_a a_s + pr_a pr_s)}$$

□

**Proof of Conclusions.** To ensure $n_{S12} > 0$, the assumption is required $pr + a_s < 2c\beta$. We obtain the relationships between the equilibriums: $ca_1{}^{*s} - ca_2{}^{*s} = -\frac{ca_s}{3pr} < 0$, $ca_1{}^{*s} - ca_1{}^{*DD} = \frac{6pr^2 - 7ca_s + 3a_a(-pr + c\beta + a_s)}{6pr}$, $ca_2{}^{*s} - ca_2{}^{*DD} = \frac{6pr^2 - 8ca_s + 3a_a(-pr + c\beta + a_s)}{6pr}$, $ca_1{}^{*DD} - ca_2{}^{*DD} = -\frac{ca_s}{2pr}$.

As to the number of subscriber and advertiser: $n_{S1}{}^{*s} - n_{S1}{}^{*DS} = n_{S2}{}^{*s} - n_{S2}{}^{*DS} = \frac{2c\beta - pr - a_s}{2pr} > 0$, $n_{S1}{}^{*DS} - n_{S2}{}^{*DS} = -\frac{ca_s^2}{3(pr^3 + pr a_a a_s)} < 0$, $n_{S1}{}^{*s} - n_{S2}{}^{*s} = -\frac{ca_s^2}{3(pr^3 + pr a_a a_s)} < 0$, $n_{S12}{}^{*DS} - n_{S12}{}^{*DD} = \frac{a_s(2pr^2 - 2(pr - c\beta)a_a - ca_s)}{2(pr^3 + pr a_a a_s)}$, $n_{S1}{}^{*s} - n_{S1}{}^{*DS} - n_{S12}{}^{*DS} = n_{S2}{}^{*s} - n_{S2}{}^{*DS} - n_{S12}{}^{*DS} = \frac{pr - 2c\beta + a_s}{2pr}$. $n_{S1}{}^{*s} - n_{S1}{}^{*DD} - n_{S12}{}^{*DD} = \frac{a_s(6pr^2 - ca_s - 3a_a(c\beta - a_s))}{6(pr^3 + pr a_a a_s)} + \frac{pr(pr - 2c\beta)}{2(pr^2 + a_a a_s)}$, $n_{S1}{}^{*s} - n_{S1}{}^{*DD} - n_{S12}{}^{*DD} = \frac{pr(pr - 2c\beta)}{2(pr^2 + a_a a_s)} + \frac{a_s(6pr^2 - 2ca_s - 3a_a(c\beta - a_s))}{6(pr^3 + pr a_a a_s)}$. $n_{A1}{}^{*s} - n_{A2}{}^{*s} = n_{A1}{}^{*DS} - n_{A2}{}^{*DS} = \frac{ca_s}{3pr^2 + 3a_a a_s} > 0$, $n_{A1}{}^{*DS} - n_{A1}{}^{*DD} - n_{A12}{}^{*DD} = \frac{3pr^2 - 3(pr - c\beta)a_a - 2ca_s}{6(pr^2 + a_a a_s)}$, $n_{A2}{}^{*DS} - n_{A2}{}^{*DD} - n_{A12}{}^{*DD} = \frac{3pr^2 - 3(pr - c\beta)a_a - ca_s}{6(pr^2 + a_a a_s)}$.

As to the profit of platform: $\pi_1{}^{*s} - \pi_2{}^{*s} = \frac{c(3pr + a_s)}{6pr} > 0$, $\pi_1{}^{*DS} - \pi_2{}^{*DS} = \frac{c(3c\beta - a_s)}{3pr} > 0$, $\pi_1{}^{*s} - \pi_1{}^{*DS} = 0$, $\pi_2{}^{*s} - \pi_2{}^{*DS} = \frac{c(2c\beta - pr - a_s)}{2pr} > 0$, $\pi_1{}^{*DD} - \pi_2{}^{*DD} = \frac{c(4cpr^2\beta + a_s(-4pr^2 + 2a_a(pr + c\beta - a_s) + 3ca_s))}{4(pr^3 + pr a_a a_s)}$. □

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
