# Peer review of "Co-Opetitive Strategy Optimization for Online Video Platforms with Multi-Homing Subscribers and Advertisers"

_jtaer, doi:10.3390/jtaer18010038_

Round 1
Reviewer 1 Report
Pls check the attachment.

Reviewer 2 Report
The paper is very important and presents actual issues. The structure of the paper is appropriate, however, some issues require explanation.
The literature survey is rather poor, so for this research paper the list of references should be increased. I would expect more publication on co-opetition strategy if authors want to focus on this issue.
In Abstract the findings concern various scenarios, however, the question is how they are combined with co-opetitive strategies. Authors are requested to explain.
In section 2.3 please, explain precisely and provide references on co-opetition as different from cooperation. Beyond that authors should explain the co-opetitive strategy and what do they mean by strategy optimization? Is it just profit maximization? If authors think just about the prodit maximization they should change the title and write directly about that, otherwise they should include additional explanation in the paper text.
Round 2
Reviewer 1 Report
Problems I raised before have been properly addressed. This paper will be more suitable for publication if following problems can be solved. I list them in order of the text:
Comment 1:
On line 16-18, more specific explanation is needed on deriving the forth conclusion in the abstract from Figure 6 (line 572-574).
Comment 2:
In total, this paper answers the question that how platforms should optimize the co-opetitive strategy considering advertisers and subscribers' single/multi-homing, as pointed out by question (3) in Introduction. While question (1) and (2) on platforms' pricing decision should be parts of this optimization. It maybe unreasonable to divide them into three parallel.
Comment 3:
On line 262, it seems inappropriate to define the transportation cost as subscriber's preference for the platform. It’s better to define it as preference for the ideal platform.
Comment 4:
Defining C2 as the broadcasting rights cost is in line with platform i's profit/cost streams as stated in their profit functions in Equation (6). If a higher c2 generally means higher content quality as you stated, it’s more appropriate to define it as the unit broadcasting rights cost while normalizing other relative cost to zero because C2Ns2 already can capture the practice that total cost increases with the number of subscribers.
Comment 5:
On characterizing multi-homing subscriber’s transportation cost like -prs(ss2-ss1) in Equation (9), more common expression should be 1. Why is the distance between them and two platforms they participate is in this paper instead of 1?
Reviewer 2 Report
Authors have provided many definitions, explanations and references.
Although the paper is valuable, I would ask authors to explain in the paper the possibilities and opportunities to verify the proposed models empirically.
Why did authors provide just theoretical considerations? Do they perceive an obstacles to verify the models in practices.
As you know empirical verification would be good enough to justify usability of the proposed models.
